



# Correcting for filter-based aerosol light absorption biases at ARM's SGP site using Photoacoustic data and Machine Learning

Joshin Kumar[1], Theo Paik[1], Nishit J. Shetty[1], Patrick Sheridan[2], Allison C. Aiken[3], Manvendra K. Dubey[3], and Rajan K. Chakrabarty[1]

[1]Center for Aerosol Science and Engineering, Department of Energy, Environmental and Chemical Engineering, Washington University in St. Louis, St. Louis, MO 63130, USA
[2]NOAA Global Monitoring Laboratory, Boulder, CO 80305, USA
[3]Earth and Environmental Sciences Division, Los Alamos National Laboratory, Los Alamos, NM 87545 USA

*Correspondence to*: Rajan K. Chakrabarty (chakrabarty@wustl.edu)

**Abstract.** Measurement of light absorption of solar radiation by aerosols is vital for assessing direct aerosol radiative forcing, which affects local and global climate. Low-cost and easy-to-operate filter-based instruments, such as the Particle Soot Absorption Photometer (PSAP) that collect aerosols on a filter and measure light attenuation through the filter are widely used to infer aerosol light absorption. However, filter-based absorption measurements are subject to artifacts which are difficult to quantify. These artifacts are associated with the presence of the filter medium and the complex interactions between the filter fibers and accumulated aerosols. Various correction algorithms have been introduced to correct for the filter-based absorption coefficient measurements toward predicting the particle-phase absorption coefficient ($B_{abs}$). Since previously-developed correction algorithms have a fixed analytical form, fundamentally, they are unable to predict particle phase absorption coefficients with a high degree of accuracy universally: different corrections are required for rural and urban sites across the world. In this study, we analyzed three months of high-resolution ambient data collected using a 3-wavelength photoacoustic spectrometer (PASS) and PSAP on the same inlet; both instruments were operated at the Department of Energy's Atmospheric Radiation Measurement (ARM) Southern Great Plains user facility in Oklahoma. We implemented the two most commonly used analytical correction algorithms, namely the Virkkula (2010) and the average of Virkkula (2010) and Ogren (2010)-Bond (1999), as well as a Random Forest Regression (RFR) machine learning algorithm to infer particle phase $B_{abs}$ values from PSAP data and estimate their accuracy in comparsion to the refernce $B_{abs}$ values measured synchronously by the PASS. The wavelength averaged Root Mean Square Error (RMSE) values of $B_{abs}$ predicted using the RFR algorithm are improved by an order of magnitude in comparison to those predicted by the Virkulla (2010) and average correction algorithms. A revised form of the Virkkula (2010) algorithm suitable for the SGP site has been proposed; however, its performance yields approximately two fold errors when compared to the RFR algorithm. To further improve the accuracy of our proposed RFR algorithm, we trained and tested the model on dataset of laboratory-generated combustion aerosols. The RFR model used as inputs the size distribution, uncorrected Tricolor Absorption Photometer (TAP)-measured $B_{abs}$, and nephelometer-measured scattering coefficient $B_{scat}$ and predicted particle-phase $B_{abs}$ values within 5% of the



reference $B_{abs}$ measured by a PASS. Machine learning approaches offers a promising path to correct for biases in long term filter-based absorption datasets and accurately quantify their variability and trends needed for robust radiative forcing

determination.

# 1 Introduction

Aerosols affect the climate through the absorption and scattering of radiation which has been the subject of intensive ongoing research (Brown et al., 2021). Aerosols are one of the most significant sources of uncertainty in climate model predictions of radiative forcing (IPCC, 2021). The U.S. Department of Energy's Atmospheric Radiation Measurement

(ARM) program was established in 1990 to collect measurements to better understand processes that affect atmospheric radiation in climate models (Stokes and Schwartz, 1994). The ARM program currently operates three heavily instrumented fixed location sites to gather atmospheric data: Southern Great Plains (SGP), North Slope of Alaska (NSA) and Eastern North Atlantic (ENA). The SGP site is the world's most comprehensive climate research facility, with extensive *in situ* and remote sensing instrument clusters deployed over about 143,000 km$^2$ centered near Lamont, Oklahoma, USA. Instruments at

the SGP site measure radiation, cloud properties, and other meteorological quantities (Sisterson et al., 2016). Light absorption by aerosols at the SGP site is measured using Manufacturer's 3-wavelength Particle Soot Absorption Photometer (PSAP) (Sheridan et al., 2001) and DMT's 3-wavelength Photoacoustic Soot Spectrometer (PASS), an extension of the 1-wavelength instrument that was deployed at Jeju island, South Korea (Flowers et al., 2010) and in Utqiagvik, Alaska (Myers et al., 2021).

The PSAP instrument infers aerosol light absorption using a low cost filter-based method by measuring transmittance through aerosol particles collected on a filter substrate. The instruments based on this method such as PSAP facilitate semi-continuous sampling of particles and produce time-averaged bulk absorption measurements (Pandey et al., 2016). Filter-based aerosol light absorption measurement instruments such as PSAP are widely used due to their low cost and operational ease, even though their accuracy suffers from "unquantifiable artifacts" such as multiple scattering which can overestimate

absorption(Bond et al., 1999; Clarke, 1982; Gorbunov et al., 2002), aerosol overloading on the filter which can underestimate absorption (Arnott et al., 1999; Weingartner et al., 2003) and the changed morphology of the deposited aerosol on the filter (Subramanian et al., 2007).

The PASS instrument was deployed at the SGP site on January 2009 followed by its decommission in October 2015 with the goal to evaluate the PSAP biases by the ARM program. The PASS is a first-principle contact-free method to measure

particle-phase aerosol light absorption coefficient ($B_{abs}$). The working principle of a PASS is described in detail in Arnott et al. (1999). Briefly, photons from a modulated laser beam are absorbed by light-absorbing aerosol particles. The absorbed energy is transmitted as heat to the surrounding air which results in a modulated pressure waves that are detected as sound waves by a microphone. The microphone can be calibrated to determine light-absorption by the particles. The measurements from a PASS are highly accurate, but they have low sensitivity (1hr average signal/noise ratio ~0.2 Mm$^{-1}$ at SGP) and long





term deployments can be expensive. PASS also have issues with liquid and/or multiphase particles, as some of the laser energy goes into the phase change associated with heating the particles rather than the producing acoustic waves.

Various correction algorithms (Bond et al., 1999; Virkkula et al., 2005; Li et al., 2020) based on a general analytical equation form, have been developed and used in climate research facilities across the world. The general form of the various previously developed correction algorithms for PSAP is summarised in Eqn. (1), where f is some function that varies

between different correction approaches and $C_0$ is a constant representing fraction of total light scattered by the particles collected on the filter.

$$B_{abs} = B_{PSAP} \times f\big(Tr(\lambda), SSA(\lambda), AAE(\lambda)\big) - C_0(\lambda) \times B_{scat} \tag{1}$$

These algorithms, however, are non-universal in applicablity and hence limited in accuracy because the fitting parameters of the transmission functions calculated in such algorithms are based on datasets of laboratory-generated aerosols which may or may not represent the diverse aerosols types in various parts of the world (Collaud Coen et al., 2010; Zuidema et al., 2018). The large variation in results of correction creates a need for a universal systematic approach for correcting filter-based measurements that is more accurate than previously stated algorithms.

In this study, we used three months of high-resolution ambient data measured by the PASS and PSAP at ARM's SGP site; we corrected for filter-based absorption measurements using Virkkula (2010) (referenced as "unrevised Virkkula" going forward), Virkkula equation with revised coefficients for the SGP site (referenced as "revised Virkkula"), the average of unrevised Virkkula and Ogren (2010) modified Bond (1999) correction (referenced as "Average"), and the Random Forest Regression (RFR), which is a supervised ensemble Machine Learning (ML) algorithm used for a wide range of classification

and regression predictive problems (Kumar and Sahu, 2021). We provide an inter-comparison of the performances of these algorithms on the sampled SGP data. Our findings show that the revised and unrevised Virkkula (2010), as well as the Average algorithms need to be significantly revised to improve their accuracy. The RFR algorithm demonstrated a high degree of accuracy in predicting $B_{abs}$ in comparison to the analytical correction forms discussed in this study.

**2 Methodology**

**2.1 Ambient data from SGP observatory**

This study used ambient ground-based aerosol data from the ARM user facility at SGP, Lamont, OK. Figure A1 presents composition data collected by the Aerodyne's Aerosol Chemical Speciation Monitor (ACSM) instrument at the SGP site over the period of ~3 months from 27th Jun to 25th Sept, 2015. We observed that organics aerosols (OA) consists of more than 60% of the mass concentration followed by sulphates, ammonium and nitrate. The summary of BC concentration at the

SGP site is shown in Fig. A2, which presents the Field Campaign Data collected using the Sunset Model 4 Semi-Continuous OC-EC Instrument from 3rd June to 27th November, 2013. The average Elemental Carbon (EC) and Organic Carbon (OC)





concentrations were found to be $0.174 \pm 0.123$ and $2.267 \pm 1.400$ ug carbon/m$^3$ air, respectively. Figure A3 illustrates the timeseries of the aerosol absorption data as measured by PSAP and PASS instruments. We observed that the average particle-phase $B_{abs}$ at the SGP site ranged from 0 to 8 Mm$^{-1}$ for most times with an average $B_{abs}$ of 1.36 Mm$^{-1}$ across all three
wavelengths.

Previous studies have measured non-refractive submicrometer aerosol concentration and the composition of its organic and inorganic constituents at the SGP site (Parworth et al., 2015; Liu et al., 2021). Across all studies, the highest mass concentration at the SGP site occurs in the winter and decreases from spring to fall. The nitrates dominate during the winters, while OA accounting for more than 60% of total non-refractory particulate matter mass concentration dominates for the rest
of the year. The $B_{abs}$ and $B_{scat}$ at 550 nm ranged from 0 to 10 Mm$^{-1}$ and 0 to 50 Mm$^{-1}$ during 2010 to 2013, respectively (Sherman et al., 2015). Also, since the site is rural, long-term transport aerosols (such as mineral dust, absorbing OA, and secondary organic aerosol – SOA) may affect local aerosol properties (Andrews et al., 2019).

In this study, high-resolution data from PASS, PSAP, nephelometer, and ACSM with sampling averaged intervals of 2sec, 1min, 1min and 30min, respectively, were collected from 27 Jun to 25 Sept 2015 in the ARM user facility at SGP (after the
High Power Green Wavelength upgrade at the SGP site in 2015). We preprocessed the data into three steps; first, we only included those timestamps where data was valid across all instruments without incorrect, suspect, and missing values. Second, we smoothed the data from all instruments into 1hr averages. Third, to compare the measurements from different instruments at the same wavelengths, we adjust the PASS-derived $B_{abs}$ and nephelometer-derived $B_{scat}$ to the PSAP's operating wavelengths. The absorption Ångström exponent (AAE) is an aerosol optical parameter used for aerosol
characterization and to extrapolate a given particle phase aerosol absorption coefficient to any wavelength of interest. The AAE and SAE values were inferred using Eqn. (2) and Eqn. (3) (Liu et al., 2018). Mean and standard deviation across time of AAE and SAE values from SGP's PASS and nephlometer data are summarized in Table A1. Since the standard deviations of AAE values for the SGP data were significantly high, time-dependent AAE and SAE values were used to extrapolate the particle phase absorption and scattering coefficients to the PSAP's operating wavelengths. The parameters $B_{abs1}$ and $B_{abs2}$ in
the Eqn. (2) and (3) are the absorption coefficients at wavelengths $\lambda_1$ and $\lambda_2$.

$$AAE = -\frac{ln(B_{abs\,1}/B_{abs\,2})}{ln(\lambda_1/\lambda_2)} \tag{2}$$

$$SAE = -\frac{ln(B_{scat\,1}/B_{scat\,2})}{ln(\lambda_1/\lambda_2)} \tag{3}$$

The extrapolation of filter-based measurements to other wavelengths using AAE is less accurate than the extrapolation of PASS measurements because filter-based measurements are inherently biased due to artifacts and its extrapolation to other wavelengths further adds on error. In order to compare the measurements from different instruments at same wavelengths,





the measured values from the particle-phase instruments - $B_{abs}$ from PASS and $B_{scat}$ from nephelometer, were extrapolated to PSAP's operating wavelengths (467, 530, and 660 nm) using inferred AAE and SAE, respectively.


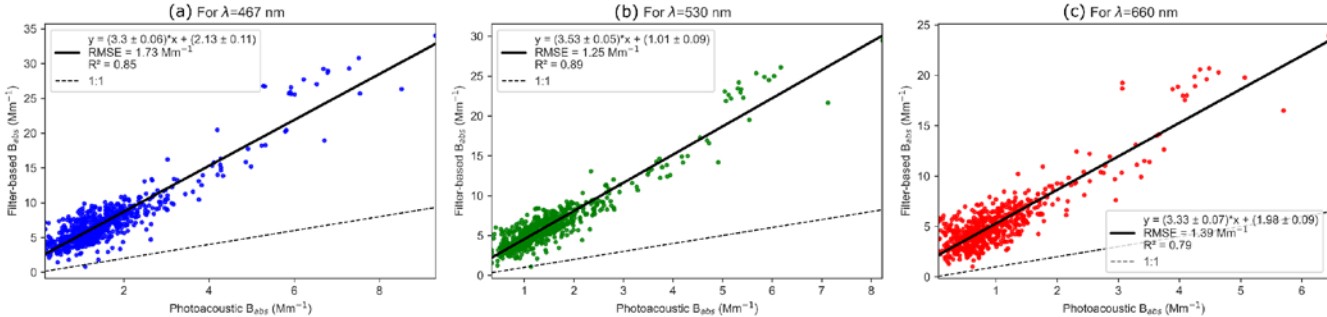

**Figure 1: Scatterplot of absorption coefficients from the PSAP and extrapolated PASS measurements corresponding to (a) 467nm, (b) 530nm and (c) 660nm wavelengths at the SGP site.**

Figure 1 presents the comparison of uncorrected filter-based absorption raw signals with the calibrated and accurate $B_{abs}$

measured using PASS. We observed that the uncorrected filter-based signals are more than 3 times greater than the accurate $B_{abs}$ measured by the PASS across all the wavelengths. Hence, at least for the SGP site, if we choose not to apply any correction algorithm on the filter-based absorption data, we can use a factor of 3 to obtain the $B_{abs}$ with a wavelength-averaged RMSE (Root Mean Square Error) of $1.45 \pm 0.2$ Mm$^{-1}$. This overestimation of the filter-based aerosol light absorption measurements is due to the enhancement of light absorption by the filter deposited aerosol due to scattering based

artifacts (Bond et al., 1999; Clarke, 1982; Gorbunov et al., 2002).

**2.2 Correction algorithms**

In order to correct for these "difficult-to-quantify" artifacts associated with the filter-based measurement of the aerosol absorption, various correction algorithms (Bond et al., 1999; Ogren, 2010; Virkkula et al., 2005; Li et al., 2020) have been introduced to predict the particle-phase absorption coefficient ($B_{abs}$) using filter-based absorption coefficient measurements.

Ogren (2010) modified Bond (1999) and Virkkula (2010) correction algorithms are widely used in global atmosphere monitoring networks such as Global Atmosphere Watch Programme (GAW) and the NOAA Federated Aerosol Network (Andrews et al., 2019). In this study we only discuss the commonly used correction algorithms on the ground sites and compared them with the proposed ML-based filter correction algorithm.

**2.2.1 Virkkula (2010) with unrevised parameters**

Virkkula et al. (2005) developed an analytical correction equation that iteratively calculates $B_{abs}$ from filter-based measurements. The transmittance correction function in the Virkkula equation was a multivariate function of the natural



logarithm of transmission and SSA as shown in Eqn. (5). The parameters in the Virkkula equation h0, h1, k0, k1 vary with wavelength. Virkkula (2010) recalculated these parameters by correcting for flowmeter calibration in Eqn. (5).

$$B_{abs} = B_{PSAP} \times (k_0 + k_1 \ln(Tr)) - s \times B_{scat} \tag{4}$$

$$B_{abs}(Virkkula\ corrected) = B_{PSAP} \times (k_0 + k_1(h_0 + h_1\omega_0)\ln(Tr)) - s \times B_{scat} \tag{5}$$

The parameters in Eqn. (4) and Eqn. (5) represent– particle phase absorption coefficient ($B_{abs}$), absorption measurement from PSAP ($B_{PSAP}$), transmission values from PSAP (Tr), particle phase scattering coefficient from Nephlometer ($B_{scat}$), single scattering albedo(SSA = $\omega_o$ = $B_{abs}/(B_{abs}+B_{scat})$)) and Virkkula parameters/constants (k0, k1, h0, h1, s).

Using these parameters of the Virkkula equation, we calculated the $B_{abs}$ values from the uncorrected filter based absorption signals. Due to the unknown values of SSA, the Virkkula equation was iteratively solved for the $B_{abs}$. The $B_{abs}$ was first calculated using the Eqs. (4) and then was used to compute the initial guess for $\omega_o$. Next, this value of $\omega_o$ was then used in Eqs. (5) to compute a more accurate value of $B_{abs}$ and this procedure was repeated until $B_{abs}$ value converged.

### 2.2.2 Virkkula (2010) with revised parameters for the SGP site

Using the reference measurements of $B_{abs}$ from the PASS at the SGP site, we refitted the parameters in the Virkkula equation (h0, h1, k0, k1) to obtain revised parameters. The fitting was implemented using the "curvefit" function from the "SciPy" Python library, which uses non-linear least squares to fit a functional equation form to given data. After fitting of optimized parameters of the Virkkula equation, we solved for the particle phase absorption coefficients using the filter-based absorption coefficients. It is important to note that the calculated revised Virkkula parameters may only be valid for the SGP site because these revised parameters were computed using the absorption data from PASS and PSAP at SGP site.

### 2.2.3 Average of unrevised Virkkula (2010) and Ogren (2010)-Bond (1999)

Bond (1999) published correction scheme for the PSAP which was updated by Ogren (2010). The Ogren (2010) modified Bond (1999) correction is applied using the Eqn. (6) to obtain corrected $B_{abs}$ value. Another correction technique that is often used by the DOE ARM community involves computing a simple arithmetic mean of Virkkula (2010) correction with unrevised parameters and the Ogren (2010)-Bond (1999) correction to obtain a average corrected $B_{abs}$ value as shown in Eqn. (7) (C Flynn et al., 2020; Zuidema et al., 2018) For brevity, going forward we will refer to this correction scheme as the "Average" correction algorithm.

$$B_{abs}(Bond/Ogren\ corrected) = B_{PSAP} \times (\frac{1}{1.5557 \times Tr + 1.0227}) - 0.0164 \times B_{scat} \tag{6}$$


$$B_{abs}(Average\ corrected) = \frac{B_{abs}(unrevised\ Virkkula\ corrected)\ +\ B_{abs}(Bond/Ogren\ corrected)}{2} \qquad (7)$$

### 2.2.4 Random Forest Regression Model

Random Forest Regression (RFR) is an ensemble supervised ML algorithm used for a wide range of classification and
regression predictive problems (Kumar and Sahu, 2021). Random forest involves constructing a large number of decision
trees with each decision tree fitted on a different subset of the training dataset (also called Bagging), in addition to selecting a
random subset of input variables at each split point in the construction of trees. Random forest is known to reduce overfitting
of data in decision trees and provide accurate predictions (Biau, 2012; Breiman, 2001). The three most essential
hyperparameters to tune the Random forest are: – 1. A number of random input variables to consider at each split point 2.
The depth of the decision trees 3. The number of decision trees in the forest. The core concept behind the Random Forest is
that it aggregates the results of many trained decision trees empirically and outputs the most optimal result.

ML algorithms perform very well on trained dataset; therefore, it is crucial to test their performance on unseen or untrained
data. We split the SGP dataset into training and testing sets in the ratio of 70:30. The training set was used to train the RFR
model, and then the testing set was used to evaluate the model's performance on the new input data that the model had not
encountered before. We trained the model using the uncorrected $B_{PSAP}$, PSAP transmission, $B_{scat}$, and total mass
concentration from ACSM as input variables and particle-phase $B_{abs}$ as the output variable. The values of the
hyperparameters used for the construction of the RFR model are: the number of features to consider while looking for the
best split = 5, the number of trees = 100, and the max_depth was such that nodes were expanded until all leaves were pure or
until all leaves contain less than two samples.

The RFR algorithm is entirely a data-driven approach to correct filter-based measurements.  The algorithm was trained on
input-output variables, which were measured by different instruments installed at the site. The instrument detection limits,
precision, and accuracy play a significant role in the training and predicting ability of the RFR algorithm. In order to gain
highly accurate predictions from the RFR algorithm on the test dataset (data that is not used while training but is used to
check the accuracy of the algorithm on unseen data), the algorithm requires good quality training data and with reasonably
large number of samples/instances  in the training dataset to ensure that the algorithm's accuracy on the unseen test dataset is
not limited by the number of samples of the training dataset on which it is trained upon. Figure A4 presents the general
workflow of ML based correction models developed in this study.





## 3 Results

### 3.1 Application of Virkkula (2010) algorithm with unrevised parameters

|  | $k_0$ | $k_1$ | $h_0$ | $h_1$ | s |
|---|---|---|---|---|---|
| **467 nm** | $0.377 \pm 0.013$ | $-0.640 \pm 0.007$ | $1.16 \pm 0.005$ | $-0.63 \pm 0.09$ | 0.015 |
| **530 nm** | $0.358 \pm 0.011$ | $-0.640 \pm 0.007$ | $1.17 \pm 0.003$ | $-0.71 \pm 0.05$ | 0.017 |
| **660 nm** | $0.352 \pm 0.013$ | $-0.674 \pm 0.006$ | $1.14 \pm 0.11$ | $-0.72 \pm 0.16$ | 0.022 |

**Table 1: Unrevised parameters as mentioned in Virkkula (2010) to be used in Virkkula algorithm (i.e., Eqn. (5)).**

215

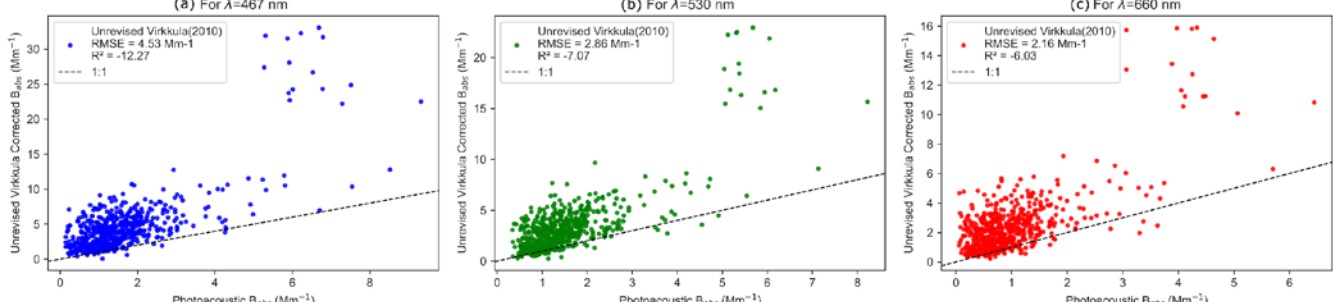

**Figure 2: Comparison between PSAP absorption coefficients, corrected for using Virkkula (2010) algorithm with unrevised coefficients, and the reference PASS absorption coefficients measured at the SGP site corresponding to (a) 467nm, (b) 530nm and (c) 660nm wavelengths.**

220    The parameters mentioned in the Virkkula (2010) as shown in Table 1 were directly used to iteratively solve for $B_{abs}$ using Eqn. (5). Figure 2 shows comparisons between the unrevised Virkkula calculated $B_{abs}$ and reference $B_{abs}$ measured using PASS. We observed that the %RMSE values (calculated over all three wavelengths as $= \Sigma_i$ ($RMSE_i$ / Mean Reference $B_{abs\_i}$)$\times 100$) which represents percentage of uncertainty for unrevised Virkkula in the calculation or predictions of $B_{abs}$ is ~240% and $R^2$ values are negative for all three wavelengths, which suggests that the unrevised Virkkula algorithm performs

225    worse than a constant prediction of mean $B_{abs}$ value.

The variance in $B_{abs}$ calculated using unrevised Virkkula is large enough to undermine the algorithm's applicability without revising the parameters/coefficients. Since fitting parameters in Virkkula (2010) were based on experimental burn data of kerosene soot and "white" ammonium sulphate aerosol, those parameters cannot be universally applied to different types of ambient aerosols (Collaud Coen et al., 2010; Zuidema et al., 2018).

230    **3.2 Application of Virkkula (2010) algorithm with revised parameters for the SGP site**

|  | $k_0$ | $k_1$ | $h_0$ | $h_1$ | s |
|---|---|---|---|---|---|
| **467 nm** | $0.292 \pm 0.008$ | $-0.011 \pm 0.008$ | $112.998 \pm 30.045$ | $-115.64 \pm 31.019$ | 0.015 |





| | | | | | |
|---|---|---|---|---|---|
| **530 nm** | $0.344 \pm 0.006$ | $0.003 \pm 0.007$ | $31.644 \pm 90.318$ | $-31.835 \pm 93.819$ | 0.017 |
| **660 nm** | $0.311 \pm 0.006$ | $0.027 \pm 0.007$ | $-60.065 \pm 8.782$ | $63.304 \pm 9.103$ | 0.022 |

**Table 2: Revised parameters for the Virkkula equation computed using SGP dataset**

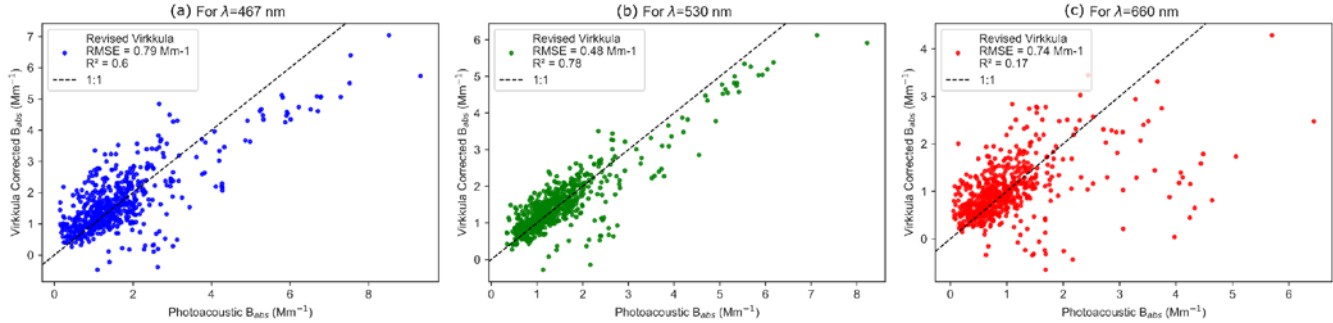

**Figure 3: Comparison between PSAP absorption coefficients, corrected for using the Virkkula algorithm with revised coefficients, and the reference PASS absorption coefficients measured at the SGP site corresponding to (a) 467nm, (b) 530nm and (c) 660nm wavelengths.**

To overcome the imprecision of the unrevised Virkkula algorithm, we fitted the Virkkula equation to the SGP data to obtain revised Virkkula parameters (i.e., k0, k1, h0, h1) shown in Table 2. The same values of s were used as mentioned in Virkkula (2010) because parameter "s" represents fraction of total light scattered which is experimentally determined by fitting to ammonium sulphate experiments (Virkkula et al., 2005). The Virkkula equation with these newly computed parameters was then used to iteratively solve for the $B_{abs}$ using Eqn. (5). Figure 3 presents a comparison of filter-based absorption corrected using the revised Virkkula algorithm and reference $B_{abs}$ measured using the PASS. We observed that the Virkkula algorithm performed comparatively well with revised parameters because the RMSE values decreased and $R^2$ values increased in comparison to unrevised Virkkula's evaluation metrics (i.e., RMSE, %RMSE and $R^2$). The results of Fig. 2 and Fig. 3 clearly imply that it is essential to revise the parameters before implementing the Virkkula equation for predicting $B_{abs}$ at each site. Since the Virkkula equation does not undertake the seasonal, source and particle size distribution as inputs, the Virkkula parameters are subject to change with these external factors too.

It is important to note that since the $B_{abs}$ predictions of revised Virkkula as shown in Fig. 3 were based on the same data that was used to calculate the Virkkula parameters, The performance of this algorithm on this data is the best that is possible. The %RMSE for the revised Virkkula predictions for the SGP data was ~57% which is less than that of unrevised Virkkula, but it still represents significant uncertainty in the calculation/prediction of $B_{abs}$. This major shortcoming of analytical fits led us to the ML approach to predict the $B_{abs}$ using filter-based measurements.

### 3.3 Application of average of unrevised Virkkula (2010) and Ogren (2010) modified Bond (1999)





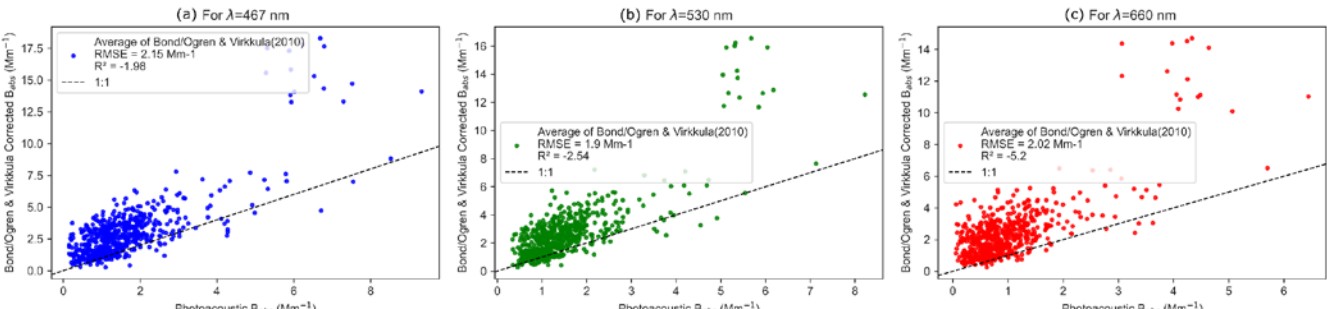


**Figure 4: Comparison between PSAP absorption coefficients, corrected for using the average of Bond/Ogren and unrevised Virkkula (2010) algorithms, and the reference PASS absorption coefficients measured at the SGP site corresponding to (a) 467nm, (b) 530nm and (c) 660nm wavelengths.**

Figure 4 presents a comparison of filter-based absorption corrected using the average of unrevised Virkkula (2010) and

Ogren (2010) modified Bond (1999), and reference $B_{abs}$ measured using the PASS. The %RMSE values for the "Average" correction are ~170% and $R^2$ are negative for all three wavelengths suggesting that the model performs worse than a constant prediction of mean $B_{abs}$ value. We observed that the "Average" correction performed better than the unrevised Virkkula but still worse than revised Virkkula algorithm. This justifies the application of "Average" algorithm at ARM sites for better accuracy when PASS-derived $B_{abs}$ values are not available to revise the parameters of Virkkula equation and using just

unrevised Virkkula algorithm yields low accuracy.

### 3.4 Application of Random Forest Regression (RFR) algorithm

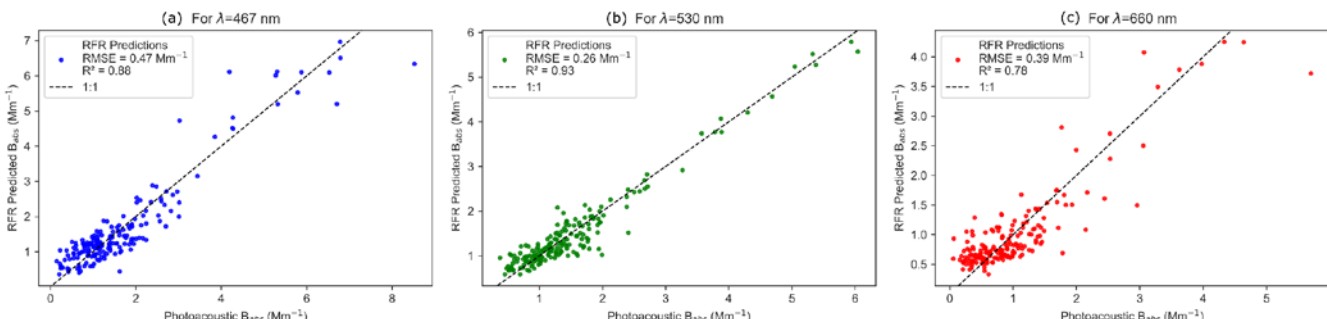

**Figure 5: Random Forest Regression, a supervised machine learning algorithm, applied to correct for PSAP absorption coefficients, and comparison of its performance with reference PASS absorption coefficients measured at the SGP site**

**corresponding to (a) 467nm, (b) 530nm and (c) 660nm wavelengths.**

We used RFR, which is a supervised ML algorithm, to correct for the filter based PSAP absorption measurements. Figure 5 presents the comparison of RFR predicted $B_{abs}$ with the reference $B_{abs}$ measured using PASS. We observed from Fig. 5 that for all three wavelengths, %RMSE values for the $B_{abs}$ predictions from RFR algorithm are ~30%, and the $R^2$ values are greater than ~0.8, which are much better than the evaluation metrics for both unrevised and revised Virkkula algorithms even

when the RFR algorithm's evaluation metrics were computed on unseen test data.





Apart from the two common correction algorithms (Ogren (2010) modified Bond (1999) and Virkkula (2010)) applied to PSAP, recent attempts were made to develop new correction algorithms (Li et al., 2020) by constructing a multivariate linear model in the general correction Eqn. (1) and including the interaction terms between AAE, SSA, and ln(Tr). It was referred as "Algorithm A" by Hanyang et al. and produced the $R^2$ values of 0.62, 0.55, and 0.43 on the PSAP's operating wavelengths

of 467nm, 528nm, and 652nm, respectively. Comparing just $R^2$ values, the RFR algorithm fares better than "Algorithm A" which is the most recent PSAP correction algorithm developed yet.

The RFR algorithm performs better than the analytical models because it empirically captures the nonlinearities and complex relationships between the input variables and $B_{abs}$, and it was trained on an extra input of total mass concentration from ACSM. It is important to note that after the eliminative pre-processing of the three months of bulk data, the number of valid

data samples that remained was relatively small for a typical ML algorithm training; we can expect that the RFR algorithm can perform even better with more extensive data.

### 3.5 Improving the accuracy of Random Forest Regression (RFR) algorithm

RFR is an ensemble supervised machine learning algorithm which builds many decision tress using the input data during the training phase and predicts the output as the mean of predictions from all of the trees. The accuracy of the RFR directly

depends on the number of different or uncorrelated trees built during the training as shown in Fig. 6. In order to produce many uncorrelated trees, we not only train the trees on different random subsets of training data (i.e., Bagging) but also choose differerent input features or variables randomly to split the nodes. Training the RFR algorithm on all the input variables which significantly affect the output variable not only enables us to increase the number of uncorrelated trees built during training but also constrains the model for accurate prediction. Hence, the accuracy of RFR to predict particle phase

$B_{abs}$ could be further improved by training the algorithm using all possible input variables that affect $B_{abs}$, such as $B_{PSAP/TAP}$, transmission, $B_{scat}$, aerosol size distribution parameters, and composition.

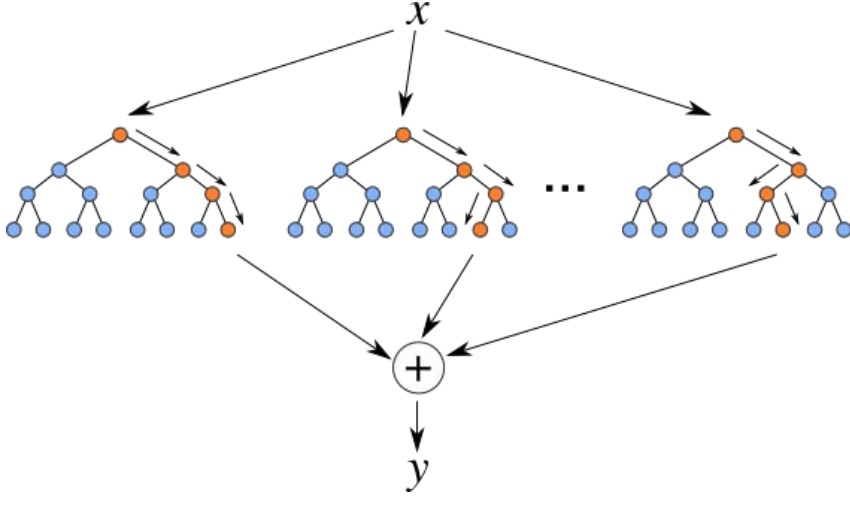





**Figure 6: Flowchart of RFR illustrating many uncorrelated trees build using random feature sampling whose average prediction is more accurate than the each of the individual trees. (Adapted from Gitconnected)**

As a proof of concept, we trained and tested the RFR algorithm on laboratory-generated published dataset of burn chamber experiments (Sumlin et al., 2018; Shetty et al., 2019; Shetty et al., 2021). The algorithm was trained using the total number concentration, geometric mean diameter, geometric standard deviation, uncorrected filter-based Tricolor Absorption Photometer (TAP) $B_{abs}$, and nephelometer $B_{scat}$ as input variables, while the output variable was the particle-phase absorption coefficient. Figure 7 presents the comparison of RFR predicted $B_{abs}$ with the reference $B_{abs}$ measured using PASS during the

burn. We observed from Fig. 7 that the RFR algorithm correctly predicted the particle-phase $B_{abs}$ within 5% (=%RMSE) of the reference $B_{abs}$. We also note that the $R^2$ values are ~1, which shows that the predictions correlate near-perfectly with the reference PASS-derived absorption values. This example demonstrates the capabilities of RFR in capturing the complex relationship between filter-based measurements and particle-phase $B_{abs}$ with the best possible accuracy.

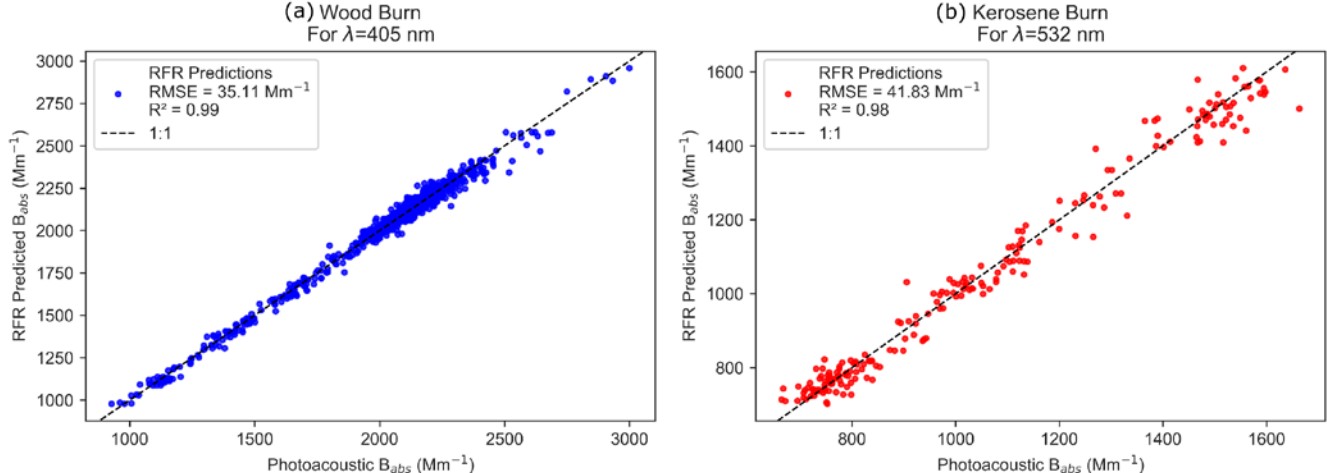

**Figure 7: An illustration of the power of Random Forest Regression (RFR) algorithm in accurately predicting particle-phase absorption coefficient when trained with a robust set of input variables. The plots show the accuracy of RFR trained TAP absorption coefficients in comparison to the reference PASS absorption coefficients corresponding to (a) 405nm and (b) 532nm for laboratory-generated combu.**

## 4 Conclusions

The uncertainties in predicting particle-phase absorption coefficients from filter-based absorption data are due to both measurement uncertainties of the instruments and the uncertainties of parameter computation while using analytical algorithms like those put forth in Virkkula (2010). Little can be done about the instruments' measurement uncertainties, originating from noise and calibration of instruments, STP correction, and flow rate uncertainties (Sherman et al., 2015). However, using ML techniques, we can avoid the uncertainties introduced from parameter computation and stiff functional

forms, which are inevitable when using algorithms with analytical forms.



We demonstrate that our RFR algorithm corrects for the PSAP filter based biases in refernce to the PASS measurements at the SGP accurately and much better than the standard Virkkula algorithm. A unique feature of the SGP site is that while there are significant monthly variations in the aerosol composition, the optical properties such as the $B_{abs}$, $B_{scat}$, and SSA are bounded in a small range with weak annual cycles. Because of this feature of the SGP site, we argue that the ML-based

correction algorithm trained in this study is scalable to other months. Furthermore, the developed correction algorithm can be applied to any climate research facility site globally, provided the seasonality information is included as an input feature to the algorithm during the training using Label Encoding method which can be used to convert categorical variable such as name of the months into numerical variable.

RFR was a ML algorithm of choice in this study because of its high accuracy even with relatively small training datasets

(Kumar and Sahu, 2021). However, if training of a large dataset is involved, other techniques such as XGBoost and neural networks could improve accuracy further than RFR. The RFR algorithm captures nonlinear dependence between variables with the highest accuracy compared to the functional analytical form correction algorithms that were previously developed. We confidently propose that ML models can produce the most accurate and fastest predictions possible of the particle phase absorption coefficients compared to any other analytical equation form algorithms, given the training data is accurate and of

reasonable size.

Major aerosol monitoring networks, such as the Interagency Monitoring of PROtected Visual Environments (IMPROVE) network and the Chemical Speciation Network (CSN) collect particle samples for measurement of UV-VIS-IR absorption coefficient. Correction scheme developed as part of this study might be applicable to infer aerosol light absorption properties for samples collected from the IMPROVE network, rural facilities and federal Class I areas. ML approaches offers

promising path to correct long term of airborne filter based absorption observations to accurately quantify their variability and trends for robust climate radiative forcing determination. Future work will be in the direction of fine tuning the RFR algorithm to accurately predicting light absorption by biomass burning aerosols from the wildfires.





**Appendix A**

|  | **Mean ± Std.** |
|---|---|
| **AAE (405-532)** | -0.132 ± 2.480 |
| **AAE (532-781)** | 2.432 ± 2.189 |
| **AAE (405-781)** | 1.366 ± 1.579 |
| **SAE (450-550)** | 1.522 ± 0.418 |
| **SAE (550-700)** | 1.781 ± 0.448 |
| **SAE (450-700)** | 1.663 ± 0.427 |

**Table A1: Mean and standard deviation of AAE and SAE values calculated for the SGP data.**

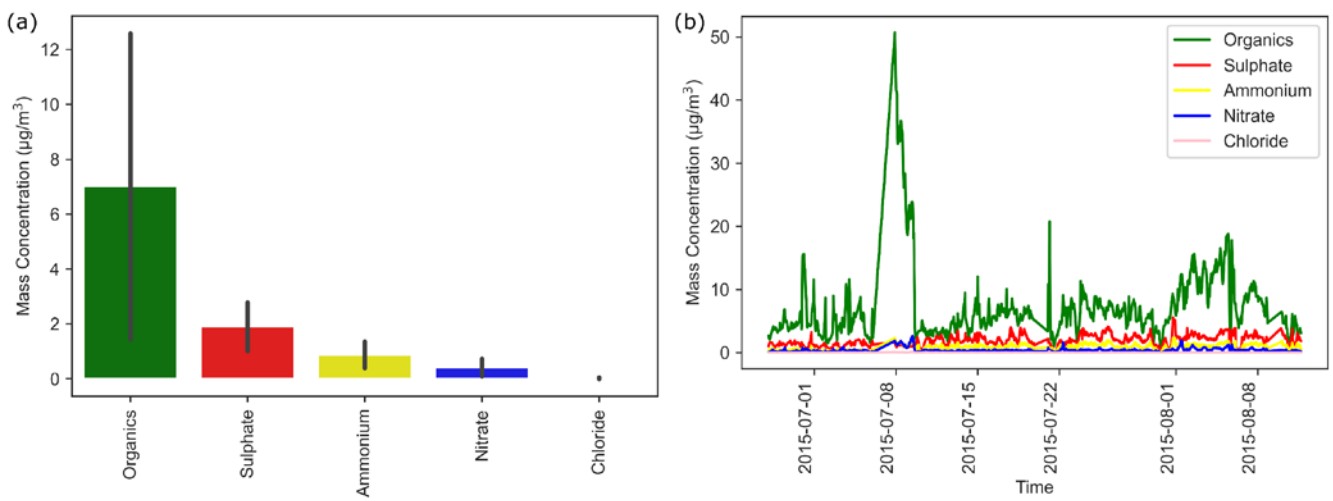


**Figure A1: Composition of the ambient ground measurement site at SGP. The error bars represent the standard deviations. (a) Mass concentrations of various species (b) Timeseries of the absolute mass concentration of particle chemical composition.**





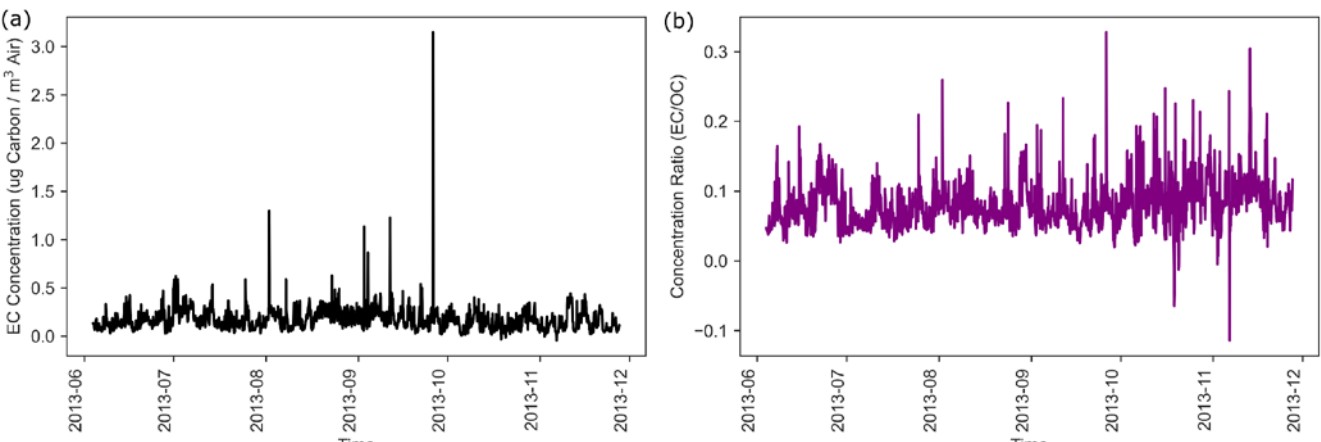

**Figure A2: (a) Timeseries of Elemental Carbon (EC) concentration. (b) Timeseries of ratio of EC and OC concentrations at SGP**
**from Jun-Nov 2013.**





**Figure A3:** Timeseries data of absorption coefficients as measured by PSAP (at 467nm, 530nm and 660nm) and PASS (at 405nm, 532nm and 781nm) instrument at the SGP observatory.





**Data Pre-Processing**
- Remove timestamps with incorrect, suspect and missing values.
- Smooth the data by grouping the timestamps into 1 hour average samples.
- Separate the input and output variables of the dataset into different dataframes.

**Test-Train Split**
- Split the pre-processed dataset into training and testing set in the ratio 70:30, respectively.

**ML Model Training**
- Train the RFR machine learning model using the input and output variables in the training dataset.

**ML Model Testing**
- Use the trained RFR model to predict output variable given the input variables of the testing dataset.

**Performance Evaluation**
- Evaluate the performance of the ML model by calculating the RMSE and $R^2$ values using the model's predicted output and true output values on the testing dataset.
- Tune the parameters of the ML model if the performance to achieve desired level of accuracy.


**Figure A4: Workflow of Machine Learning based correction model developed and used in this study.**





**Code availability**

https://github.com/joshinkumar/Filter-correction-ML-code.git

**Data availability**

Atmospheric Radiation Measurement (ARM) user facility. 2009. Photoacoustic Soot Spectrometer (AOSPASS3W). 2015-06-27 to 2015-09-25, Southern Great Plains (SGP) Central Facility, Lamont, OK (C1). Compiled by A. Aiken. ARM Data Center. Data set accessed 2021-12-17 at http://dx.doi.org/10.5439/1190011.

Atmospheric Radiation Measurement (ARM) user facility. 2011. Particle Soot Absorption Photometer (AOSPSAP3W). 2015-06-27 to 2017-09-25, Southern Great Plains (SGP) Central Facility, Lamont, OK (C1). Compiled by A. Koontz and S. Springston. ARM Data Center. Data set accessed 2021-12-17 at http://dx.doi.org/10.5439/1333829.

Atmospheric Radiation Measurement (ARM) user facility. 2011. Nephelometer (AOSNEPHDRY). 2015-06-27 to 2015-09-25, Southern Great Plains (SGP) Central Facility, Lamont, OK (C1). Compiled by A. Koontz and J. Uin. ARM Data Center. Data set accessed 2021-12-17 at http://dx.doi.org/10.5439/1258791.

Atmospheric Radiation Measurement (ARM) user facility. 2010. ACSM, corrected for composition-dependent collection efficiency (ACSMCDCE). 2015-06-27 to 2015-09-25, Southern Great Plains (SGP) Central Facility, Lamont, OK (C1). Compiled by M. Zawadowicz and J. Howie. ARM Data Center. Data set accessed 2021-12-17 at http://dx.doi.org/10.5439/1763029.

Field Campaign Data: Semi-Continuous OCEC SGP 2013:
https://adc.arm.gov/discovery/#/results/id::6561_ocec_microchem_scocec_aerosol_blkcarbonconc?showDetails=true

Laboratory generated wood and keroscene burn dataset:
https://github.com/joshinkumar/Filter-correction-ML-code/blob/main/Lab%20Burn%20Dataset.zip

**Financial support**

This research has been primarily supported by the US Department of Energy (grant no. DE-SC0021011). The laboratory experiments of the study were partially supported by the National Science Foundation (grant no. AGS-1926817).



405 **Competing interests**. The authors declare that they have no conflict of interest

**Author contributions.** RKC conceived of the study and its design. JK performed the data analysis, developed and implemented the models, and led the preparation of the manuscript. MKD and ACA collected PASS dataset at the SGP site. TP performed the laboratory experiments. RKC, NJS, and PS provided guidance and supervision for carrying out the 410 research tasks, interpretation of results, and contributed to the preparation of the manuscript. All authors were involved in the editing and proofreading of the manuscript.

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
