# Peer review of "Correcting for filter-based aerosol light absorption biases at ARM's SGP site using Photoacoustic data and Machine Learning"

_Atmospheric Measurement Techniques, 2022_

## Referee Comment (RC2)

Kumar et al.: **Correcting for filter-based aerosol light absorption biases at ARM's SGP site using Photoacoustic data and Machine Learning**, Atmos. Meas. Tech. Discuss. https://doi.org/10.5194/amt-2022-42, in review, 2022.

**Review**

**General**

The paper presents the use of a supervised ensemble Machine Learning (ML) algorithm for improving the constants in an algorithm used for calculating absorption coefficients from PSAP data and for calculating a new algorithm for the same purpose. The method interesting and useful since it can improve the accuracy of absorption meassurements. The paper is definitely worth publishing in AMT but I do have some suggestions and several questions that should be answered before that. They are all in the detailed comments and questions below.

**Detailed comments and questions**

L 45: Why not replace "Manufacturer's" with "Radiance Research"?

L66 - 79: You should also cite Müller et al. (AMT, 7, 4049–4070, https://doi.org/10.5194/amt-7-4049-2014, 2014) and note that it is based on much more rigorous theory of radiative transfer through the filter.

L72, Eq. (1): First, I suggest you don't present the equation in the introduction, it would be much more logical to present it in section" 2.2 Correction algorithms". However, where ever you present it, you should define exactly what you mean by $B_{PSAP}$. Is it the $B_{PSAP}$ presented in Eq. (3) of Bond et al. (1999)? If it is, you should keep in mind that it already includes one loading-correction function f. In other words, what is your "uncorrected filter-based absorption" all over the paper? The RR 3wl PSAP firmware calculates automatically absorption coefficients corrected with the Bond et al. correction excluding the scattering correction. With user-defined constants. So is that what you think that is the "uncorrected $B_{PSAP}$"? If so, that is not quite correct. The uncorrected $B_{abs}$ shoud be $B_{PSAP}$ divided by the loading correction function f that the PSAP firmware uses. Explain in more detail. And further, if you really have used the $B_{PSAP}$ calculated directly by the PSAP and assumed that it is the "uncorrected absorption" then you have to recalculate everything! I hope not. Recheck that!

L86-88: "Our findings show … ". I suggest you move this to the conclusions. These are all results of the whole study. In the intro you should present the goals and in the conclusions the main result.

Section 2.1 You should write something about the inlets, flows, cutoffs and size ranges for the different instruments. These are not just for fun, they are important info to try to evaluate the sources of the differences of the absorption coefficients from the different instruments.

In this section you should also tell, which filter material you used in the PSAP. That is important because the constants in the algorithms depend on the filter material.

L110-111: In this preprocessing, did you divide the $B_{PSAP}$ with the f(Tr) that is automatically calculated by the instrument firmware?

L116-117: The AAE from the PASS data in Table A1 are somewhat suspicious. Especially the ones that have the wl 532 included. The fact that AAE(405-532) < -0 and AAE(532-781) > 2 suggest that $B_{ABS}$(532) is overestimated. Then this would have important implications to the factors presented in Table 2. Discuss this.

Further about Table A1. I strongly suggest you add more in formation in it. You have not presented anywhere the descriptive statistics (ave, std, some percentiles) of the aerosol optical properties during the campaign. That would be very important because it would show the range in which your results are applicable. Present $B_{abs}$, $B_{sca}$, SSA, AAE from the different instrument and algorithm combinations. Especially AAE is important, from the different algorithms. Peope use them for source apportionment. And in my opinion this table is important, it would deserve to be in the main text.

L271-275: So, this is not just an adjustment of the factors in Virkkula (2010). Do I understand right: the RFR has given as an output an equation that was used for calculating the absorption coefficients. If so, you should present the equation so that other people can use it also! What are the parameters the new function depends on?

L301-307: Please give the full functions so that other people can use and test them. What are the derived constants for wood burning and kerosene burning smoke?

Fig. A2a: You do have also EC concentration data! I suggest you use those data also, not just in this plot. Excluding the obvious outliers in the data shown in A2a, how do the the absorption coefficients with the different methods correlate with EC? From the linear regressions you would get mass absorption coefficients. What would be the derived $MAC(\lambda)$? Anything close to published ones? These info would not be just for fun, they would be an additional support for the values derived from the different algorithms.

---

## Author Comment (AC1)

*The authors would like to thank the reviewer1 for all their suggestions. We have addressed the comments and provide a point-by-point response to the recommendations made by the reviewer1 below. The reviewer1 comments are in black, and our responses are in red.*

The manuscript "Correcting for filter-based aerosol light absorption biases at ARM's SGP site using Photoacoustic data and Machine Learning" by J. Kumar et al. shows that a random forest tree machine learning algorithm can correct particle absorption coefficients measured with filter-based instruments. The analysis was performed for a specific measurement site, but the study itself can be used as a blueprint for testing ML algorithms for other stations with various aerosol types.

The manuscript addresses a relevant topic and falls within the scope of AMT. It is well written and the conclusions are sound. The reviewer recommends the manuscript for publication after considering the following minor comments.

The authors would like to thank the reviewer for providing comprehensive and insightful suggestions. We have made appropriate changes to the manuscript as detailed in the reviewer's specific comments.

**Specific comments:**

Page 1, line 18: The reviewer does not fully agree with the chosen explanation or wording why filter-based instruments have problems in predicting the particle absorption coefficient. The reviewer believes that the reason for the limitations is not that fixed analytical (*) forms were chosen, but rather that there are hidden influencing parameters. There is also no algorithm that takes into account all known influencing parameters, e.g. Nakayama et al. (2010) present a correction for particle penetration depth, the restricted two-stream method (Mueller et al., 2014) takes into account particle asymmetry but not particle size. It should not be concluded that solvers with fixed analytical functions are generally unable to predict particle absorption coefficients with high accuracy.

(*The reviewer means that iterative solvers for fixed parameterised functions are also included in the class of fixed analytic functions. )

The author agrees with the reviewer. The references provided are cited and the statement is now correctly expressed in the updated version of the manuscript as follows – *"However, the inability of these algorithms to incorporate in their formulations the complex matrix of influencing parameters such as particle asymmetry parameter, particle size, and particle penetration depth, results in prediction of particle-phase absorption coefficients with relatively low accuracy."*

Page 1, line 31: Does the RFR model use the particle size distribution as input? Cf. line 195, where it says that the total mass concentration is used as input. What does total mass concentration mean? Is the cumulative mass on a PSAP filter spot meant?

The RFR model is applied to two different cases in this paper:

1.  The first application is on the SGP data where Input variables given to the RFR model are = [uncorrected BPSAP, PSAP transmission, Bscat, total mass concentration from ACSM which is

simply the sum of mass concentrations of Organics, Sulphate, Ammonium, Nitrate, and Chloride obtained from ACSM (Refer Fig. A2(a)) in now updated manuscript].

2. The second application is on data obtained from Lab-generated aerosols where the following Input variables to the RFR model are given = [Aerosol number size distribution parameters (N, u_g, sigma_g) obtained from SMPS, uncorrected (TAP)-measured Babs, and nephelometer-measured scattering coefficient Bscat]. The output variable of RFR is the same in both cases, i.e., the corrected particle-phase absorption coefficient.

This is now updated and clearly described in the abstract (Page1; Lines 25-30) and Section 2.2.4(Line 215-220) to avoid any confusion. "*For the application on the SGP dataset, the RFR model was trained with an uncorrected absorption coefficient derived from PSAP ($B_{abs\_uncorrected\_PSAP}$), PSAP transmission (Tr), scattering coefficient from nephelometer ($B_{scat}$), and total mass concentration obtained from the sum of ACSM measured concentrations of various species as input variables and particle-phase $B_{abs}$ as the output variable.*"

Line 59: Can PASS be considered a first principle method? A few lines later the authors describe the problem with liquid or multiphase particles, which is a fundamental problem of the method?

We agree with the reviewer. This statement is now corrected to "*The PASS is a contact-free method to measure particle-phase aerosol light absorption coefficient ($B_{abs}$)*".

Line 98: Do the authors mean the absorption coefficients or the uncorrected absorption coefficients?

This statement is now updated to "*Figure A4 illustrates the timeseries of the aerosol absorption data as derived from PSAP ($B_{abs\_uncorrected\_PSAP}$) and PASS ($B_{abs}$) instruments.*" to avoid any further confusion.

Figure A4 (Previously Figure A3) illustrates the time series of the absorption coefficients as derived from the PSAP {($B_{abs\_uncorrected\_PSAP}$)} and measured by the PASS. The PASS measurements are more accurate; hence, they are used as a reference to compare corrected filter-based absorption coefficients in this study}. Figure A4 just presents all the data from PSAP and PASS so that the reader can be assured that the data correlates well across different instruments and is of good quality.

In Figures A3 and 1, the axis label and caption should indicate whether corrected or uncorrected absorption coefficients are shown.

Figure A4 (Previously Figure A3) represents the raw data used in this study for the SGP site from PASS {as measured} and PSAP {($B_{abs\_uncorrected\_PSAP}$) = uncorrected value derived after undo the automatic B1999 correction implemented by PSAP firmware}.

The caption of Figure A4 is now updated to reflect this. We also made changes in the manuscript text by defining "*the uncorrected aerosol absorption data as derived from PSAP ($B_{abs\_uncorrected\_PSAP}$)*" in equation form to avoid any confusion. We have now updated the terminology in the manuscript. Now, $B_{abs\_PSAP}$ means the absorption coefficient output from the PSAP firmware which includes the automatic B1999 correction and $B_{abs\_uncorrected\_PSAP}$ refers to the back calculated uncorrected PSAP-based absorption coefficient before the B1999 correction that is auto applied by the Radiance Research's PSAP firmware.

$$B_{abs\_uncorrected\_PSAP} = \frac{A_{PSAP}}{Q_{PSAP}\Delta t}ln\left(\frac{I(t)}{I(t+\Delta t)}\right) = \frac{B_{abs\_PSAP}}{f(Tr)} = \frac{B_{abs\_PSAP}}{\left(\frac{1}{1.317 \times Tr\ +\ 0.866}\right)} \quad (1)$$

**General comments to chapter 2:**

Because of the known artefact due to light scattering particles, it would be informative if the authors presented single scatter albedos.

Thank you for this suggestion. To present the SGP data better, we now present descriptive statistics of all the instruments and the derived optical parameters like SSA, AAE and SAE obtained from the SGP data in the Appendix section.

[Figure]

*Figure A1: Summary of the SGP dataset. The boxplots of raw measurement data are shown as obtained from various instruments used in this study (a) PASS (b) PSAP{ B_{abs_uncorrected_PSAP} } and (c) Nephelometer. The boxplots of parameters dervied from the raw data are also shown (d) AAE (e) SAE and (f) SSA. The green line is the median of the data. The bottom line of box is 25% percentile of data and top line of box is 75% percentile of data, therefore, the box represents the middle 50% of all the datapoints which is the core of the data.*

Why was the Virkkula (2010) correction revised but not the Ogren (2010)-Bond (1999) correction?

Thank you for pointing this out. We now show the performance of both Ogren-Bond and Virkkula separately before showing the "Average" performance of both algorithms. The reason for not revising Bond-Ogren is now also included in the manuscript as follows:- "*The %RMSE for this (unrevised Bond-Ogren) algorithm on the SGP data is ~312% which is almost the same as the %RMSE of unrevised Virkkula (2010). Since the general equation form of Ogren (2010) modified Bond (1999) is similar to that of Virkkula (2010) and both the unrevised versions of algorithms perform with similar accuracy, hence, the improvement in accuracy of Ogren (2010) modified Bond (1999) with revised coefficients can be expected to be very similar to that in the case of Virkkula.*"

[Figure]

Figure 4: Comparison between PSAP absorption coefficients, corrected for using Bond-Ogren correction algorithm, and the reference PASS absorption coefficients measured at the SGP site corresponding to (a) 467nm, (b) 530nm and (c) 660nm wavelengths.

[Figure]

Figure 2: Comparison between PSAP absorption coefficients, corrected for using Virkkula (2010) algorithm with unrevised coefficients, and the reference PASS absorption coefficients measured at the SGP site corresponding to (a) 467nm, (b) 530nm and (c) 660nm wavelengths.

Figure 5: It seems that there are fewer data points in Figure 5 than in Figures 3 and 4. Is the split of the data into training and test data sets the only reason?

Yes, the sole reason for less number of points in Figure 6 and Figure 8 {previously Fig 5 and Fig 7} (*application of RFR on SGP data and RFR on lab-generated aerosol data*) is due to the fact that we are only showing RFR's performance on the test data (30% of randomly shuffled data) which the RFR did not encounter during training and was used only for prediction and calculating RMSE. We did this to show that RFR has lower RMSE scores even on unseen data as compared to the traditional algorithms which yield high RMSE values even when predicting on the same data that they used to train/revised coefficients.

Line 303: Does it have any influence that the laboratory dataset was taken with a TAP photometer and the data from the SPG site was taken with a PSAP photometer?

As mentioned earlier in detail the RFR is applied to two different cases in this paper, and both the RFR models were trained independently. The first RFR model was trained and tested with SGP data and the other RFR model was trained and tested on lab-generated aerosol data, to further establish RFR as a more accurate correction algorithm on aerosols from various sources. Since these models are trained independently, they do not influence from each other's input data.

Appendix A4: check sentence: "Tune the parameters of the ML model if the performance to achieve desired level of accuracy."

Thank you for pointing out, all grammatical mistakes found in the manuscript are now corrected.

**References:**

Nakayama, T., et al. (2010). "Size-dependent correction factors for absorption measurements using filter-based photometers: PSAP and COSMOS." Journal of Aerosol Science 41(4): 333-343.

Mueller, T., et al. (2014). "Constrained two-stream algorithm for calculating aerosol light absorption coefficient from the Particle Soot Absorption Photometer." Atmos. Meas. Tech. 7: 4049-4070.

---

## Author Comment (AC2)

*The authors would like to thank the reviewer2 for all their suggestions. We have addressed the comments and provide a point-by-point response to the recommendations made by the reviewer2 below. The reviewer2 comments are in black, and our responses are in* *red**.*

Kumar et al.: **Correcting for filter-based aerosol light absorption biases at ARM's SGP site using Photoacoustic data and Machine Learning**, Atmos. Meas. Tech. Discuss. https://doi.org/10.5194/amt-2022-42, in review, 2022.

**Review**

**General**

The paper presents the use of a supervised ensemble Machine Learning (ML) algorithm for improving the constants in an algorithm used for calculating absorption coefficients from PSAP data and for calculating a new algorithm for the same purpose. The method interesting and useful since it can improve the accuracy of absorption meassurements. The paper is definitely worth publishing in AMT but I do have some suggestions and several questions that should be answered before that. They are all in the detailed comments and questions below.

The authors would like to thank the reviewer for providing comprehensive and insightful suggestions. We have made appropriate changes to the manuscript as detailed in the reviewer's specific comments.

**Detailed comments and questions**

L 45: Why not replace "Manufacturer's" with "Radiance Research"?
This statement is now replaced in the updated version of the manuscript as follows: "*Light absorption data by aerosols at the SGP site is collected using Radiance Research's 3-wavelength Particle Soot Absorption Photometer (PSAP)…*"

L66 - 79: You should also cite Müller et al. (AMT, 7, 4049–4070, https://doi.org/10.5194/amt-7-4049-2014, 2014) and note that it is based on much more rigorous theory of radiative transfer through the filter.
This paper is now cited in the manuscript.

L72, Eq. (1): First, I suggest you don't present the equation in the introduction, it would be much more logical to present it in section" 2.2 Correction algorithms". However, where ever you present it, you should define exactly what you mean by $B_{PSAP}$. Is it the $B_{PSAP}$ presented in Eq. (3) of Bond et al. (1999)? If it is, you should keep in mind that it already includes one loading-correction function f. In other words, what is your "uncorrected filter-based absorption" all over the paper? The RR 3wl PSAP firmware calculates automatically absorption coefficients corrected with the Bond et al. correction excluding the scattering correction. With user-defined constants. So is that what you think that is the "uncorrected $B_{PSAP}$"? If so, that is not quite correct. The uncorrected $B_{abs}$ shoud be $B_{PSAP}$ divided by the loading correction function f that the PSAP firmware uses. Explain in more detail. And further, if you really have used the BPSAP calculated directly by the PSAP and assumed that it is the "uncorrected absorption" then you have to recalculate everything! I hope not. Recheck that!

Thank you for correctly pointing this out. As you advised, we have now used the uncorrected $B_{abs}$ as $B_{PSAP}$ divided by the loading correction function f(Tr). We have also recalculated everything and updated the numerical values in the text, tables, and figures. This causes a small non-uniform scaling of values as compared to the previous wrong "uncorrected Babs" values that we were using; hence it changes the numerical result values slightly and but does not affect the overall conclusions of the study. However, we have corrected and updated all the Figures in the manuscript affected due to this additional preprocessing. We also made changes in the manuscript text by

defining "*the uncorrected aerosol absorption data as derived from PSAP ($B_{abs\_uncorrected\_PSAP}$)*" in equation form. We have now updated the terminology in the manuscript. Now, $B_{abs\_PSAP}$ means the absorption coefficient output from the PSAP firmware which includes the automatic B1999 correction and $B_{abs\_uncorrected\_PSAP}$ means the back calculated uncorrected PSAP-based absorption coefficient without the B1999 correction.

$$B_{abs\_uncorrected\_PSAP} = \frac{A_{PSAP}}{Q_{PSAP}\Delta t} ln\left(\frac{I(t)}{I(t+\Delta t)}\right) = \frac{B_{abs\_PSAP}}{f(Tr)} = \frac{B_{abs\_PSAP}}{\left(\frac{1}{1.317 \times Tr \ + \ 0.866}\right)} \qquad (1)$$

L86-88: "Our findings show … ". I suggest you move this to the conclusions. These are all results of the whole study. In the intro you should present the goals and in the conclusions the main result.

This paragraph is now shifted to conclusions and updated.

Section 2.1 You should write something about the inlets, flows, cutoffs and size ranges for the different instruments. These are not just for fun, they are important info to try to evaluate the sources of the differences of the absorption coefficients from the different instruments.

This information is now included in the manuscript in lines from 110-125 in Section 2.1.

In this section you should also tell, which filter material you used in the PSAP. That is important because the constants in the algorithms depend on the filter material.

This information is now updated in Section 2.1. "*The PSAP has been operated by ARM (and many others in the global community) for almost 25 years with the same filter media, Pallflex E70-2075W, which is composed of quartz fibers on a cellulose backing.  All published corrections factors were developed and measured using the Pallflex E70 media.*"

L110-111: In this preprocessing, did you divide the $B_{PSAP}$ with the f(Tr) that is automatically calculated by the instrument firmware?

As mentioned in an earlier response, we took care of this preprocessing now and updated all the figures and affected text in the manuscript.

L116-117: The AAE from the PASS data in Table A1 are somewhat suspicious. Especially the ones that have the wl 532 included. The fact that AAE(405-532) < -0 and AAE(532-781) > 2 suggest that $B_{ABS}$(532) is overestimated. Then this would have important implications to the factors presented in Table 2. Discuss this.

To present the SGP data better, we now present Figure A1 in updated manuscript which includes descriptive statistics of measurements from all the instruments and the derived optical parameters like SSA, AAE and SAE obtained from the SGP data in the Appendix section.

In Table A1, the average AAE calculated using 405nm and 532nm wavelength was negative because, as it can be seen in Figure A1(d), there are many negative outliers for AAE_PASS_405_532 which was pulling the average below zero. Please note that the medians {green line} of all the AAEs calculated are positive.

We have now stated this in the manuscript as well – "*From figure A1(a), (d) and figure A4, however, we suspect that either the newly installed 532nm PASS laser could be slightly overestimating absorption, or that the old 405nm and 781nm lasers could be slightly underestimating absorption compared to their true values.*"

We agree that due to unavailability of highly accurate aerosol particle-phase light absorption data, we had to resort to assuming available PASS instrument's measurements as ground truth for absorption, which limits the accuracy of the revised Virkkula parameters shown in Table2.

[Figure]

*Figure A1: Summary of the SGP dataset. The boxplots of raw measurement data are shown as obtained from various instruments used in this study (a) PASS (b) PSAP{ $B_{abs\_uncorrected\_PSAP}$ } and (c) Nephelometer. The boxplots of parameters dervied from the raw data are also shown (d) AAE (e) SAE and (f) SSA. The green line is the median of the data. The bottom line of box is 25% percentile of data and top line of box is 75% percentile of data, therefore, the box represents the middle 50% of all the datapoints which is the core of the data.*

Further about Table A1. I strongly suggest you add more information in it. You have not presented anywhere the descriptive statistics (ave, std, some percentiles) of the aerosol optical properties during the campaign. That would be very important because it would show the range in which your results are applicable. Present $B_{abs}$, $B_{sca}$, SSA, AAE from the different instrument and algorithm combinations. Especially AAE is important, from the different algorithms. Peope use them for source apportionment. And in my opinion this table is important, it would deserve to be in the main text.

Thank you for this suggestion. To present the SGP data better, we now present Figure A1 in the manuscript which includes descriptive statistics of measurements from all the instruments and the derived optical parameters like SSA, AAE and SAE obtained from the SGP data in the Appendix section.

L271-275: So, this is not just an adjustment of the factors in Virkkula (2010). Do I understand right: the RFR has given as an output an equation that was used for calculating the absorption coefficents. If so, you should present the equation so that other people can use it also! What are the parameters the new function depends on?

Yes, this is not an adjustment of factors in Virkkula (2010) here. However, since, RFR is a machine learning algorithm it cannot be expressed in a simple equation form. Machine Learning models can only predict the output using the input data provided to them once they are trained on both input and output data (training dataset).

L301-307: Please give the full functions so that other people can use and test them. What are the derived constants for wood burning and kerosene burning smoke?

We can only train machine learning models like RFR and test their output accuracy on unseen input data. Roughly speaking, tree-based algorithms can be thought of as a series of many if-else statements nested together, hence, we cannot obtain a simple looking mathematical equation out of a trained model. However, for practical applications, ML models can be part of post-processing of data at various sites to produce highly accurate outputs possible.

Fig. A2a: You do have also EC concentration data! I suggest you use those data also, not just in this plot. Excluding the obvious outliers in the data shown in A2a, how do the the absorption coefficients with the different methods correlate with EC? From the linear regressions you would get mass absorption coefficients. What would be the derived MAC(λ)? Anything close to published ones? These info would not be just for fun, they would be an additional support for the values derived from the different algorithms.

Unfortunately, we do not have EC concentration data for the period of focus in this paper (27th June-25th September). This period was chosen because it gave us access to good quality data, parallely, for all the required instruments for this study (PSAP, PASS, NEPH, ACSM), and a new high powered green laser PASS was upgraded at the SGP site in 2015 [1]. The only reason behind Fig. A3a (previously Fig. A2a) being present in the appendix of manuscript is to give the reader a glimpse of the SGP site's typical EC concentrations, since most of the results are using SGP data. The availability of the data can be cross-checked at the ARM's Data Discovery site [2].

References:
1) https://github.com/joshinkumar/Filter-correction-ML-code/blob/main/Dubey%20Poster%202015.pdf
2) https://adc.arm.gov/discovery/#/

---

## Author Comment (AC3)

*The authors would like to thank the reviewer3 for all their suggestions. We have addressed the comments and provide a point-by-point response to the recommendations made by the reviewer3 below. The reviewer3 comments are in black, and our responses are in red.*

**General comments**

Kumar et al. describe a very interesting comparison of "traditional" PSAP correction algorithms with a new machine learning algorithm. The work is important and can contribute significantly to the painful post-processing of the filter-photometer data in general. Filter photometers are used widely and in very different environments, so an algorithm that reduces the bias with no or very little assumptions is most welcome.

The reviewer will take the unusual action of sticking to general comments (under several titles) and have only two specific ones. The importance of the results is unfortunately influenced by the hastiness and shallowness of the writing. There is a definite lack of attention to detail. The paper should be heavily revised and reviewed afterwards. It definitely deserves publication – the improvement in the regressions between the PSAP and PASS is impressive.

The authors would like to thank the reviewer for providing comprehensive and insightful suggestions. We have made appropriate changes to the manuscript as detailed in the reviewer's specific comments.

**Terminology and parameters**

What is B_PSAP?

$B_{PSAP}$ was supposed to be uncorrected absorption coefficient from PSAP. I had earlier mistaken and used the absorption coefficient from PSAP directly which were auto corrected by Bond's correction by the PSAP firmware. Thank you for pointing this out. We have now updated the terminology in the manuscript. Now, $B_{abs\_PSAP}$ means the absorption coefficient output from the PSAP firmware which includes the automatic B1999 correction and $B_{abs\_uncorrected\_PSAP}$ means the back calculated uncorrected PSAP-based absorption coefficient without the B1999 correction.

$$B_{abs\_uncorrected\_PSAP} = \frac{A_{PSAP}}{Q_{PSAP}\Delta t} ln\left(\frac{I(t)}{I(t+\Delta t)}\right) = \frac{B_{abs\_PSAP}}{f(Tr)} = \frac{B_{abs\_PSAP}}{\left(\frac{1}{1.317 \times Tr + 0.866}\right)} \quad (*)$$

We have also updated all the affected Figures of the manuscript after doing this additional pre-processing to obtain raw absorption data from PSAP.

What are "uncorrected filter-based absorption raw signals"?

We have now updated this in the manuscript to "*Figure 1 presents the comparison of uncorrected filter-based absorption coefficients with the calibrated, particle-phase $B_{abs}$ measured using PASS*". We meant to say uncorrected PSAP's absorption coefficient (without the loading correction which is automatically applied by Radiance Research's PSAP).

The statement (line 138) that "this overestimation…" – by filter pohotometers, "… is due to the enhancement of light absorption by the filter deposited aerosol due to scattering based artifacts" is misleading. The enhancement is due to scattering of light by the filter fibers. Part of the reduction of light intensity below the sample is due to scattering (away from the forward direction). This is correctly summarized in Eq. 1, but not here. The separation of these two effects is artificial (Mueller et al., 2010). This should be discussed, especially in light of the superiority of the RFR algorithm.

Thank you for pointing this misleading statement. This has been corrected as per your recommendation "*This overestimation of the filter-based aerosol light absorption measurements is due to the scattering of light away from the forward direction by the filter fibers and due to the changed morphology of the deposited aerosol on the filter (Subramanian et al., 2007; Bond et al., 1999; Clarke, 1982; Gorbunov et al., 2002).*"

The authors should use notation and naming of the parameters consistent with Ogren et al (2017). I suspect that the "uncorrected filter-based absorption raw signals" are in fact the attenuation coefficient. If this is so, please use this parameter. The "raw signals" could be interpreted as raw intensity signals measured by the photodiode… Please be precise.

Thank you for pointing this out. This has been corrected in the manuscript as mentioned earlier. Now, $B_{abs\_PSAP}$ means the absorption coefficient output from the PSAP firmware which includes the automatic B1999 correction and $B_{abs\_uncorrected\_PSAP}$ means {raw data} the back calculated uncorrected PSAP-based absorption coefficient without the B1999 correction.

Also, the absorption coefficient is derived, not measured, by filter photometers. The authors correctly state this, but then relapse into claiming that the absorption coefficient is "measured" by PSAP.

These misleading statements related to PSAP has been updated to "derived" along with clear representation of symbols defined in the manuscript.

What is the reason for including Eq 4 before Eq 5? It is never referenced.

As mentioned in Virkkula (2005), Eqn. (4) was used for first fitting for k0 and k1 and then using those values Eqn. (5) was used to further fit for obtaining h0 and h1. Eqn. (4) was shown only as an intermediate step taken in calculations as per Virkkula (2005). It is referenced in the next paragraph, in lines 185-190.

**Measurements**

Start by explicitly stating the period under investigation. This sets the stage. It seems there is 6.5 years of collocated PASS and PSAP data, yet the authors use only 3 months?!

The period under investigation (27[th] June to 25[th] Sept, 2015) is stated in Section 2.1, where the SGP site is introduced. The prime reason for choosing this short period were: 1) PASS at SGP site was

upgraded with a high-power green laser and deployed at SGP in 2015 {See Reference}, so we wanted to analyse the data after 2015. 2) This period contained parallel good quality data across all the instruments (PSAP, PASS, NEPH, ACSM) we required for this study to apply and compare various filter-correction algorithms. The availability of the data can be cross-checked at the ARM's Data Discovery site.

Which filter was used in the PSAP? Pall, Azumi, anything else… - see for example Ogren et al., 2017 for the difference in regression slope.

Thank you for pointing this out. This information is now mentioned in the manuscript as follows: *"The PSAP has been operated by ARM (and many others in the global community) for almost 25 years with the same filter media, Pallflex E70-2075W, which is composed of quartz fibers on a cellulose backing. All published corrections factors were developed and measured using the Pallflex E70 media."*

Was the inlet dried? Was there a cutoff? How was the flow split? Were the ACSM measurements performed in the same size fraction? Please provide all relevant details!

Thank you for pointing this out. These details are now updated in Section 2.1 of the manuscript.

The authors state that SSA is not available. This is probably not true, as the ARM web page:

https://www.arm.gov/capabilities/instruments?category=aerosol&type=armobs&site=sgp

states that there is a neph installed at SGP from 4 Oct 2010 onwards. Include the scattering data everywhere as it will improve the corrections, especially the Virkkula 2010 corrected algorithm.

The period under investigation (27$^{th}$ June to 25$^{th}$ Sept, 2015) is stated in Section 2.1, where the SGP site is introduced. The prime reason for choosing this small period were: 1) PASS at SGP site was upgraded with a high-power green laser and deployed at SGP in 2015. 2) This period contained parallel good quality data across all the instruments (PSAP, PASS, NEPH, ACSM) that we required for this study. The availability of the data can be cross-checked at the ARM's Data Discovery site.

We do use scattering data from Nephelometer in this study when applying Virkkula and other algorithms.

References:

1) https://github.com/joshinkumar/Filter-correction-ML-code/blob/main/Dubey%20Poster%202015.pdf
2) https://adc.arm.gov/discovery/#/

SSA is fundamental in terms of the overestimation of derived absorption using filter-photometers (Weingartner et al., 2003; Virkkula et al., 2015; Yus-Díez et al., 2021). The analysis of the performance (see below on the comment which parameter to use) of the algorithms as a function of SSA should be investigated.

Since machine Learning algorithms require independent variables as inputs, The RFR model in this study is trained using the uncorrected absorption coefficient derived from PSAP ($B_{abs\_uncorrected\_PSAP}$), PSAP transmission (Tr), scattering coefficient from nephlometer ($B_{scat}$), and total mass concentration from ACSM as input variables and particle-phase $B_{abs}$ as the output variable. As both absorption and scattering parameters (although from different phases) are passed as input in the model the RFR is capable of roughly inferring SSA (or similar output influencing dependent parameter) which is a combination of particle phase absorption and scattering coefficients and its influence over model's output.

We agree that it would be interesting to see the performance of RFR in periods of different SSA. It is very likely that the performance of ML models remains the same {given training data extensively covers the periods of different SSA for that site} as the model is capable of roughly inferring SSA since it is not an independent input (depends only upon absorption and scattering which we are already providing as inputs) and hence does not impart additional unique information to the model. We thank the reviewer for this comment and we aim to find this relationship between RFR model's performance and SSA in our future work with FIREX-data where the period of investigation would be longer to work with.

Similarly, ACSM measurements are supposed to be available from 18 Nov 2010 onwards. The selectio of only 3 months makes the huge OM event starting on 2015 07 07 very important when looking at the "average" picture – it heavily skews the data, if using averages, especially since the PASS and PSAP were not working consistently during this period. The ACSM measurements could be used to a higher degree in the interpretation of the results (see also below).

All the timeseries data plotted is averaged over 1hr for noise reduction for all instruments used mentioned in this study. Proof that PASS, PSAP and ACSM were working consistently is that during OM event is that there is high absorption (from both PASS and PSAP) and high Organic concentration (from ACSM) at the same time. As you recommended, we are using ACSM measurements as input to RFR for better predictions of particle phase absorption from raw uncorrected PSAP data. OM event actually provides the opportunity to the RFR model to learn from the correlations between the inputs and outputs which makes it robust for predicting Babs from filter data in future for new input data.

What is the relevance Fig A2, showing, among other things, negative OC? The period is 2 years prior to the measurement campaign.

Fig A3 {previously Fig A2} is only attached to show that at SGP site the variation in BC concentration is quite low and constant. We did not find the EC/OC data during our period of investigation in 2015 for SGP. The availability of the data can be cross-checked at the ARM's Data Discovery site.

The authors use negative AAE derived from the blue/green wavelength pair for inter/extrapolation. Such values are highly unusual and require major attention. There must be an error somewhere, since other AAE values seem closer to what one would expect. The OM event probably has a huge influence on AAE. What happened, a large fire?

We thank the reviewer for careful observation of this data, we sincerely apologize for the

misunderstanding caused. To present the SGP data better, we now present Figure A1 in updated manuscript which includes descriptive statistics of measurements from all the instruments and the derived optical parameters like SSA, AAE and SAE obtained from the SGP data in the Appendix section. In Table A1, the average AAE calculated using 405nm and 532nm wavelength was negative because, as it can be seen in Figure A1(d), there are many negative outliers for AAE_PASS_405_532 which was pulling the average below zero. Please note that the medians {green line} of all the AAEs calculated are positive. We have now stated this in the updated manuscript as well – *"Observing Figure A1(a), (d) and Figure A4, we suspect that either the newly installed 532nm PASS at the site{See Reference} in 2015 could be slightly overestimating absorption, or the existing old 405nm PASS and 781nm PASS could be slightly underestimating absorption as compared to their true values.*

[Figure]

*Figure A1: Summary of the SGP dataset. The boxplots of raw measurement data are shown as obtained from various instruments used in this study (a) PASS (b) PSAP{ $B_{abs\_uncorrected\_PSAP}$ } and (c) Nephelometer. The boxplots of parameters dervied from the raw data are also shown (d) AAE (e) SAE and (f) SSA. The green line is the median of the data. The bottom line of box is 25% percentile of data and top line of box is 75% percentile of data, therefore, the box represents the middle 50% of all the datapoints which is the core of the data.*

**Data processing and Algorithms**

How were the period with incorrect, suspect, and missing values identified? What were the criteria?

We only meant to say that only those timestamps were used which did not had "NaN" values in any of the instruments (PASS, PSAP, NEPH, ACSM). Apart from this, in the PASS data, those few timestamps which have absorption coefficient negative and more that 15Mm-1 were removed after observing the timeseries plot since they were obvious outliers for SGP site, this is also consistent with literature that the SGP site has a small absorption range from around 0 to 10Mm-1 (Sherman et al., 2015). Similarly, in the ACSM data, the timestamps with negative concentrations were removed from further processing. Finally, this "without-outliers" data from each of the instruments was first averaged for 1hr intervals and then concatenated parallelly with time.

This statement is now updated to avoid confusion as follows – *"we only included those timestamps where data was valid across all instruments without incorrect (e.g. Negative absorption coefficients),*

*suspect (e.g. PASS measurements > 15Mm$^{-1}$ at the SGP site), and missing values (e.g. Missing timestamps corresponding to parallel instrument measurements)".*

How was averaging performed? Was the Springston and Sedlacek (2007) algorithm used? It should be at least investigated, there is an interesting example by Backman et al. (2017) which could be followed.

We appreciate the reviewer pointing us to this past interesting research on this topic, However, in the context of this study, the less the data is processed the better it is for training the RFR which is a machine learning-based algorithm that infer patterns in data. Therefore, we prefer to not pre-process the training data of this study with previously published algorithms whose limitations are hard to estimate.

The reason for using the average of Virkkula (2010) and Ogren (2010)-Bond (1999) is described only briefly. It would help to treat each separately and then show the average (which is used in processing of the data). One expects a comment also on why is this paper better than Arnott et al. (2005).

We have now updated the manuscript and treated both algorithms separately before averaging them together. The ARM sites use the combined average of the Virkkula and Ogren-Bond (C Flynn et al., 2020), hence, the average was shown to compare its accuracy with other algorithms including the proposed RFR algorithm. Since Arnott el al. (2005) also assumes a similar fixed general equation format to that of Virkkula (2010) and various comparison studies already exist between these two algorithms which justify their comparable accuracies. Our focus was on comparing the most commonly used algorithms at the ARM sites (Virkkula, Bond, Ogren) with the proposed RFR algorithm.

[Figure]

Figure 4: Comparison between PSAP absorption coefficients, corrected for using Bond-Ogren correction algorithm, and the reference PASS absorption coefficients measured at the SGP site corresponding to (a) 467nm, (b) 530nm and (c) 660nm wavelengths.

How is the training/testing split 70/30 determined?

The division of data is into training set (on which the model is trained on and is exposed to the data's inputs and outputs) is and test set (in which only inputs are given to the trained model and model outputs obtained are compared with the actual output data) is human defined. Generally accepted ratio in the machine learning community are 80/20 OR 70/30. More training data does not always

means a significantly better model since the accuracy improves marginally once the model is trained on sufficient data to infer the underlying patterns.

More details need to be provided. Why is the learning period twice as long as the test one? What happens if the periods are extended (6.5 years of data!)? Is the 70/30 split pre-defined by a human or is this some sort of a Monte Carlo sampling?

The amount of test data does not change the model in any way, It is just used to compare the predictions from the model with actual output data that the model did not encounter during training. As you correctly inferred, It is pre-defined by the person building the model, Ideally, the splitting of the data available into the training set is done such that the model's performance on the test set does not show any improvement with increasing the size of the training data set. It is also important to note that that data was randomly shuffled before the splitting into 70% training and 30% test set.

The RFR method is empiricistic. It would be of great interest to check its performance in periods of different SSA… to see what are the real parameter of interest (see the back-scattering coefficient and SSA in Virkkula et al., 2015)?

We agree that it would be interesting to see the performance of RFR in periods of different SSA. It is very likely that the performance remains the same as the model is capable of roughly inferring SSA since it is not an independent input (depends only upon absorption and scattering which we are already providing as inputs) and hence does not impart additional unique information to the model. We thank the reviewer for this comment and we aim to find this relationship between RFR model's performance and SSA in our future work with FIREX-data where the period of investigation would be longer to work with.

The reviewer is not sure that RMSE is the correct parameter to estimate the performance of the correction algorithms. It assumes the error only in the PSAP measurements. While this is true for "bias" assuming that PASS is the "absolute truth" (see above), but it is not true for experimental noise.

We agree with the reviewer that PASS also have error in its measurements. However, since we hourly average the PASS absorption coefficients the noise is significantly reduced, and the obtained average value is the best available estimate of "absolute truth" in this study to compare the accuracy of various filter-correction algorithms with. However, we do recognize this limitation of RMSE and hence also provide each plot (*of corrected values vs PASS*) with $R^2$ (*Coefficient of determination*) which gives additional information on the correlation between corrected values and reference PASS values and is consistent with RMSE scores across various algorithms applied.

What is the cause of RMSE wavelength variability? Noise? This is algorithm independent – green regressions are always best.

Since PASS at the SGP site was upgraded with a new high-power green laser and deployed at SGP in 2015, The RMSE wavelength variability could be because of the PASS instrument's lasers at different

wavelengths might have different noise/errors causing small wavelength dependence. It could be possible that the newly installed green laser is more accurate than the others, yielding good quality data which when compared with corrected filter-based absorption coefficients gives lower RMSE values.

**Results**

Why is the number of points in the regression on Fig 5 (RFR) lower that for other algorithms?

The sole reason for the less number of points in Fig 6 (RFR) {Previously Fig 5} is due to the fact that we are only showing the test data (30% of randomly shuffled data) which the RFR did not encounter during training and was used only for prediction and calculating RMSE. We did this to show that RFR has lower RMSE scores even on unseen data as compared to the traditional algorithms which yield high RMSE values even when predicting on the same data that they were train/revised upon.

**Miscellaneous**

I am curious: could you derive Virkkula parameters with the RFR algorithm?

Unfortunately, this is the prime limitation of machine learning that the model cannot be represented in terms of simple mathematical equations. Roughly speaking, RFR can be thought of as a sequence of "if and else" statements nested together to form a tree-like structure which is difficult to write in an equation form. However, for practical applications, ML models can be part of post-processing of data at various sites to produce highly accurate outputs possible.

The laboratory experiment (Section 3.5) is very different but interesting. There should be more experimental detail.

Since the focus of this paper was correcting and comparing filter-based absorption coefficients with PASS data, the detailed information on the experiments available in the cited references in Section 3.5. These references are previous papers from our lab group describing the laboratory experiment on generating by burning fuels are cited: "(Sumlin et al., 2018; Shetty et al., 2019; Shetty et al., 2021)".

**Specific comments**

Please spell check the manuscript!

All the typos found in the manuscript are now corrected.

Please use the dates in global format (18 Nov 2010), so that our colleagues from outside the Americas will understand them without ambiguity.

All the time-series figures in the manuscript are now updated to global date format.

**References**

Arnott, W. P., Hamasha, K., Moosmüller, H., Sheridan, P. J., and Ogren, J. A.: Towards aerosol light-absorption measurements with a 7-wavelength Aethalometer: Evaluation with a photoacoustic instrument and 3-wavelength nephelometer, Aerosol Sci. Technol., 39, 17-29, 10.1080/027868290901972, 2005.

Backman, J., Schmeisser, L., Virkkula, A., Ogren, J. A., Asmi, E., Starkweather, S., Sharma, S., Eleftheriadis, K., Uttal, T., Jefferson, A., Bergin, M., Makshtas, A., Tunved, P., and Fiebig, M.: On Aethalometer measurement uncertainties and an instrument correction factor for the Arctic, Atmos. Meas. Tech., 10, 5039–5062, https://doi.org/10.5194/amt-10-5039-2017, 2017.

Müller, T., Virkkula, A., and Ogren, J. A.: Constrained two-stream algorithm for calculating aerosol light absorption coefficient from the Particle Soot Absorption Photometer, Atmos. Meas. Tech., 7, 4049–4070, https://doi.org/10.5194/amt-7-4049-2014, 2014.

Ogren, J. A., Wendell, J., Andrews, E., and Sheridan, P. J.: Continuous light absorption photometer for long-term studies, Atmos. Meas. Tech., 10, 4805–4818, https://doi.org/10.5194/amt-10-4805-2017, 2017.

Springston, S. R. and Sedlacek, A. J.: Noise characteristics of an instrumental particle absorbance technique, Aerosol Sci. Tech., 41,1110–1116, https://doi.org/10.1080/02786820701777457, 2007.

Virkkula, A.: Correction of the calibration of the 3-wavelength Particle Soot Absorption Photometer (3λ PSAP), Aerosol Science and Technology, 44, 706-712, 2010.

Virkkula, A., Chi, X., Ding, A., Shen, Y., Nie, W., Qi, X., Zheng, L., Huang, X., Xie, Y., Wang, J., Petäjä, T., and Kulmala, M.: On the interpretation of the loading correction of the aethalometer, Atmos. Meas. Tech., 8, 4415–4427, https://doi.org/10.5194/amt-8-4415-2015, 2015.

Weingartner, E., Saathoff, H., Schnaiter, M., Streit, N., Bitnar, B., and Baltensperger, U.: Absorption of light by soot particles: determination of the absorption coefficient by means of Aethalometers, J. Aerosol Sci, 34, 1445-1463, 10.1016/S0021-8502(03)00359-8, 2003.

Yus-Díez, J., Bernardoni, V., MoÄ□nik, G., Alastuey, A., Ciniglia, D., IvanÄ□iÄ□, M., Querol, X., Perez, N., Reche, C., Rigler, M., Vecchi, R., Valentini, S., and Pandolfi, M.: Determination of the multiple-scattering correction factor and its cross-sensitivity to scattering and wavelength dependence for different AE33 Aethalometer filter tapes: a multi-instrumental approach, Atmos. Meas. Tech., 14, 6335–6355, https://doi.org/10.5194/amt-14-6335-2021, 2021.